# Recent Advances in the Molecular Biology of Systemic Mastocytosis: Implications for Diagnosis, Prognosis, and Therapy

**DOI:** 10.3390/ijms21113987

**Published:** 2020-06-02

**Authors:** Margherita Martelli, Cecilia Monaldi, Sara De Santis, Samantha Bruno, Manuela Mancini, Michele Cavo, Simona Soverini

**Affiliations:** Department of Experimental, Diagnostic and Specialty Medicine, Hematology/Oncology “L. e A. Seràgnoli”, University of Bologna, 40138 Bologna, Italy; cecilia.monaldi@studio.unibo.it (C.M.); sara.desantis3@studio.unibo.it (S.D.S.); samantha.bruno2@unibo.it (S.B.); manuela.mancini6@unibo.it (M.M.); michele.cavo@unibo.it (M.C.); simona.soverini@unibo.it (S.S.)

**Keywords:** systemic mastocytosis, KIT tyrosine kinase, allele burden, next-generation sequencing, SETD2 histone methyltransferase

## Abstract

In recent years, molecular characterization and management of patients with systemic mastocytosis (SM) have greatly benefited from the application of advanced technologies. Highly sensitive and accurate assays for *KIT* D816V mutation detection and quantification have allowed the switch to non-invasive peripheral blood testing for patient screening; allele burden has prognostic implications and may be used to monitor therapeutic efficacy. Progress in genetic profiling of *KIT*, together with the use of next-generation sequencing panels for the characterization of associated gene mutations, have allowed the stratification of patients into three subgroups differing in terms of pathogenesis and prognosis: (i) patients with mast cell-restricted *KIT* D816V; (ii) patients with multilineage *KIT* D816V-involvement; (iii) patients with “multi-mutated disease”. Thanks to these findings, new prognostic scoring systems combining clinical and molecular data have been developed. Finally, non-genetic SETD2 histone methyltransferase loss of function has recently been identified in advanced SM. Assessment of SETD2 protein levels and activity might provide prognostic information and has opened new research avenues exploring alternative targeted therapeutic strategies. This review discusses how progress in recent years has rapidly complemented previous knowledge improving the molecular characterization of SM, and how this has the potential to impact on patient diagnosis and management.

## 1. Introduction

Mastocytosis is a rare neoplasm characterized by a wide range of clinical manifestations due to excessive proliferation and an increased accumulation of morphologically and immunophenotypically abnormal clonal mast cells (MCs) in different organs and tissues. MCs are immune system effector cells that, either directly or via cross-talk with other cells, participate in both innate and acquired immunity. They are often compared to a double-edged sword: on one hand, MCs function as mediators of type I allergic and inflammatory immune responses; on the other hand, they have a key protective role against bacteria, parasites, and some environmental threats like reptile and insect venom. MCs derive from bone marrow (BM) CD34+/CD117+ pluripotent progenitor cells that enter the bloodstream as committed precursors (mast-cell progenitors, MCPs). MCPs then reach various peripheral tissues (mainly those serving as barriers against environmental antigens, like the skin and the mucosal surfaces of the eyes and the gastrointestinal and respiratory tracts) where they ultimately undergo differentiation and maturation. MCs were first observed and described in 1879 by Paul Ehrlich [1]. He coined the term “Mastzellen” hypothesizing that they might have a nutritional function because of their granules, containing substances that reacted metachromatically with aniline dyes. Now we know that these granules (more than thousand per cell) contain pre-formed biologically active mediators like serotonin, histamine, heparin, tryptase, and chymase. Additionally, MCs may secrete mediators that are de novo synthesized upon cell stimulation (Platelet activating factor, Prostaglandin D2, Leukotriene B4 and Leukotriene D4), as well as a wide array of cytokines (Interkeukin (IL)-1, IL-3, IL-5, IL-8, IL-10, Granulocyte-Macrophage Colony-Stimulating Factor, Tumor Necrosis Factor-α, Trasforming growth factor-β (TGF-β), and Vascular Endothelial Growth Factor that, taken together, exert pro-inflammatory, chemotactic and immunomodulatory functions in response to well-defined signals. 

The proliferation and survival of MCs occur through the binding of stem cell factor (SCF) to the extracellular domain of the KIT receptor. In addition to SCF, MC growth and survival is modulated by locally expressed cytokines such as IL-3, IL-4, IL-9, and Tβ [2]. In systemic mastocytosis (SM), proliferation and activation are uncontrolled and amplified, and this translates into a variable symptomatology in relation to the extent of MC infiltration to the organ(s) involved and to the eventual association with other diseases of hematological origin. 

According to the 2016 World Health Organization (WHO) classification, SM includes five major subtypes: indolent SM (ISM), smoldering SM (SSM), SM with an associated clonal hematopoietic non-MC disease (SM-AHN), aggressive SM (ASM), and MC leukemia (MCL) [3,4]. SM-AHN, ASM, and MCL are also grouped under the definition of “advanced SM” (AdvSM).

Due to the difficulty in recognizing symptoms and the rarity of this disease, the incidence and prevalence of SM are likely to be underestimated. Diagnosis of SM can be established when at least the major and one minor criterion, or, three minor criteria are found [3,4]. The major diagnostic criterion is the detection, at the level of BM or other extra-cutaneous organs, of dense multifocal infiltrates consisting of cohesively aggregated MCs (>15 MCs). These are often atypical and with a reduced granule content. Minor criteria include the following: a) >25% of all MCs are atypical on BM smears or are spindle-shaped in infiltrates detected on sections of visceral organs; b) MCs in BM or peripheral blood (PB) or another extracutaneous organ express CD2 and/or CD25; c) baseline serum tryptase level is >20 ng/mL; d) a *KIT* point mutation at codon 816 is detected in the BM or another extracutaneous organ [3,4];. the SM subtypes are defined by the presence/absence of the so-called ‘‘B’‘ and ‘‘C’‘ findings. “B” findings are a measure of disease burden: >30% of MCs in the BM; serum tryptase level >200 ng/mL; signs of dysplasia or proliferation not diagnostic of another hematological malignancy, hepatomegaly or splenomegaly without impairment of organ function. “C” findings are related to organ dysfunction and include: cytopenia due to massive BM infiltration; impaired liver function with ascites and/or portal hypertension; hypersplenism; osteolytic bone lesions and/or pathological fractures; malabsorption and weight loss. ISM meets the diagnostic criteria but has less than two B findings; SSM has two or more B findings (but no C findings); ASM has one or more C findings [4]. ISM is the most common subtype and has a relative benign prognosis [5,6], although 5–10% of ISM patients have been reported to progress to SSM or to AdvSM [7,8,9]. SSM, which is associated with an increased symptom burden, has an inferior survival compared to ISM, and a much higher transformation rate (15–20% for transformation to AdvSM or acute leukemia) [5,10]. The prognosis of AdvSM is quite poor [5,6].

Whereas treatment of ISM is generally palliative and simply aimed at controlling the secretion of the MC mediators or alleviating their effects, treatment of AdvSM remains a critical issue. Very recently, the first KIT tyrosine kinase inhibitor—midostaurin—has been approved. However, a proportion of patients do not respond to or relapse on midostaurin, which underlines the need for additional treatment options.

From a molecular point of view, the last decade has witnessed significant advancements, facilitated by the advent of novel high-throughput technologies, that have contributed to a better understanding of the pathogenesis of the disease, refined diagnostication, and prognostication and identified novel putative therapeutic targets. Here we provide a review of how these advances have integrated previous knowledge and how an improved molecular characterization of the disease is likely to impact on better patient management.

## 2. KIT Activating Mutations as the Main Disease Driver

The human proto-oncogene c-*KIT*, homolog of the feline sarcoma retrovirus oncogene *v-KIT*, was first described by Yarden et al. in 1987 [11], when it was found to encode a transmembrane tyrosine kinase receptor structurally related to the macrophage and platelet-derived growth factor receptors, capable of activating mitogenic signals when stimulated [12]. Soon after, the steel or stem cell factor (SCF) was identified as the KIT receptor ligand [13,14] which was found to play a role in the development of melanocytes, hematopoietic stem cells, germ cells, and interstitial cells of Cajal [13]. 

*KIT* is located on the long arm of chromosome 4 (4q11–4q13) and contains 21 exons that overall span 80 kb of DNA [15]. The promoter of the gene contains a functional binding site for the transcription factor MITF that is used to regulate its expression in MCs [16], in addition, the levels of mRNA and protein are positively or negatively modulated by cytokines locally expressed, as well as by SCF itself [17]. 

*KIT* encodes a 976 amino acid protein of 145 kd molecular weight, a member of the type III tyrosine-kinase receptor superfamily composed of an extracellular domain (ECD) characterized by 5 Ig-like domains that contain a ligand-binding site for SCF (second and third Ig-like domain) and a dimerization site (fourth domain) [18]. ECD is linked to a cytoplasmic region by a transmembrane domain (TMD). The cytoplasmic region consists of a juxtamembrane domain (JMD) and a bipartite tyrosine-kinase domain (TKD) where the ATP binding site and the phosphotransferase domain (PTD) are separated by a kinase-insert [18]. 

KIT is involved in the differentiation of myeloid and lymphoid lines and is down-regulated in mature cells, except for MCs. Binding to SCF leads to dimerization and autophosphorylation of the receptor at tyrosine residues serving as docking sites for signal transduction molecules. The transduction process involves multiple different pathways such as PI3 kinase, Src family kinase, Ras-Erk pathway, and JAK/STAT, resulting in cell proliferation, survival, and migration. Alternative mRNA splicing is responsible for the production of four different protein isoforms that differ from the presence/absence of the ‘‘GNNK’‘ motif in the JMD or of a serine in the TKD, that seem to have a different tumorigenic potential due to the different ability to induce signal transduction [19,20,21,22].

The abnormal kinase activity of KIT has been documented in different human malignancies. In addition to SM, it has been reported in germ cell tumors, melanomas, gastrointestinal stromal tumors (GIST) and acute myeloid leukemias (AML; primarily those with inv(16) or t(8;21)) [23,24,25,26]. 

The most common activating mutations occur either in the activation loop (‘‘enzymatic pocket mutations’’) or in the JMD (‘’regulatory type mutations”). The first group of mutations affects the structure of the enzymatic pocket, the second group impair the regulation of the kinase activity of the protein [27,28,29,30]. This classification has important therapeutic implications since tyrosine kinase inhibitors can be predicted to be effective against the second type of mutations, but their binding is likely to be negatively affected by mutations belonging to the first group [28]. Activating mutations of *KIT* play a crucial role in the pathogenesis of SM by enabling the proliferation and survival of abnormal MCs in affected tissues. The first identification of gain-of-function mutations occurred in the HMC-1 human MCL cell line, which was found to harbor a constitutively activated KIT receptor. Sequencing of KIT cDNA revealed two heterozygous point mutations resulting in a valine to glycine amino acid substitution at codon 560 (V560G) and in an aspartate to valine substitution at codon 816 (D816V) [31]. Kitayama et al. expressed the murine equivalents of D816V and V560G mutations in Ba/F3 cell lines, finding that they lead to growth factor-independent proliferation in vitro and tumorigenicity when injected into nude mice [32]. The vast majority (>80%) of SM patients harbor the D816V mutation. Variants like D816Y, D816F, D816H, and D816I have also occasionally been found. Mutations at other codons have been reported, particularly in some MCL cases [33]. Figure 1 shows the map of all reported mutations. In children (where mastocytosis is essentially cutaneous and tends to regress spontaneously at puberty), mutations have mainly been mapped in the ECD [34]. Single nucleotide polymorphisms (SNPs), like the M541L, have also been found in pediatric and adult mastocytosis [35]. 

## 3. Mutant KIT D816V Receptor Induces Abnormal Signaling Pathways

The D816V gain of function mutation leads to a conformational change in the PTD of the KIT receptor, which entails its constitutive activation [36]. Several studies have engaged in the comparison of downstream signaling pathways activated by oncogenic KIT D816V versus wild-type KIT. Both qualitative and quantitative differences in signaling potential have been observed. In particular, it has been shown that *KIT* D816V induces a range of abnormal signaling pathways based on its subcellular localization and/or its substrate specificity [37]. Abnormal signaling pathways that are activated by the KIT D816V receptor may explain the aggressiveness of SM and may highlight novel potential therapeutic targets. 

Among the key mediators of oncogenic KIT D816V signaling, phosphatidylinositol 3 kinase (PI3K) [38], signal transducer and activator of transcription 5 (STAT 5) [39,40], nuclear factor kappa B (NF-kB) [41], mammalian target of rapamycin complex 2 (mTORC2) [42], and protein kinase C delta (PKCδ) [43] have been reported. 

PI3K regulates cell metabolism, growth, migration, survival, and angiogenesis. It has been proven that PI3K is more strongly phosphorylated in immortalized murine progenitor cells (MIHCs) expressing KIT D816V compared to MIHCs expressing wild-type KIT [38]. The same study also showed that not all signal pathways downstream of PI3K are constitutively active in MIHC-D816V cells, but that Jun N-terminal kinase 1 (JNK1) and 2 (JNK2) were strongly phosphorylated. Thus, KIT D816V constitutively associates with PI3K and may promote cellular transformation through the activation of JNK family members [38]. Several agents aimed at targeting the PI3K pathway are in clinical trials [44]. 

KIT D816V also induces the constitutive activation of STAT proteins, which are intracellular transcription factors involved in cell proliferation, differentiation, and survival. In particular, it has been demonstrated that STAT5 is activated in *KIT* D816V neoplastic MCs of SM patients. STAT5, located in the cytoplasm, forms a signaling complex with PI3K, resulting in activation of AKT [39]. Constitutive activation of the STAT5/PI3K/AKT signaling cascade leads to the KIT D816V dependent growth and survival of neoplastic MCs [39,40]. In addition, STAT5 induces the expression of oncostatin M, which is a modulator of the BM microenvironment whose increased expression causes enhanced angiogenesis, thickened trabeculae, and fibrosis that are all characteristic of SM [45]. STAT5 inhibitors are still actively studied [39,40,46]. Another signaling pathway represented by FAK/TIAM1/RAC1/PAK1 regulates the nuclear translocation of STAT5 active in *KIT* D816V mutant SM patients [47]. Along similar lines, Martin et al. [48] studied the VAV1/RAC1/PAK signaling pathway downstream of the KIT D816V mutated receptor. Both authors have shown that FAK and PAK inhibitors lead to an increased suppression of neoplastic MCs derived from SM patients [47,48]. Like STAT5, another transcription factor called MITF plays an essential role in proliferation and is highly expressed in MCs of *KIT* D816V-mutated patients [49]. Lee et al. demonstrated that KIT D816V represses the expression of two miRNA (miR-539 and miR-381) normally involved in down-regulation of MITF in MCs.

Tanaka et al. have observed that in HMC-1 cells carrying the D816V and V560G KIT mutations there is a constitutive activation of NF-kB [41]. NF-kB is a dimeric transcription factor that plays a primary role in the regulation of immune response and serves as a pivotal mediator of inflammatory responses, but also for cell growth, survival, and proliferation. Its activation, mediated by KIT receptor autophosphorylation, is essential for neoplastic MC proliferation [41]. It has also been shown that IMD-0354, an NF-kB inhibitor, may block its DNA-binding activity and, as a result, completely suppress proliferation in HMC-1 cells [41].

Gabillot-Carre et al. detected the constitutive activation of the mTORC signaling pathway in HMC-1 cells bearing the D816V mutation [42]. This complex is involved in the regulation of cell growth, protein synthesis, and cell cycle progression. This provides support for the potential use of mTOR inhibitors like rapamycin in the treatment of aggressive SM patients expressing D816V-mutated *KIT* [42]. Subsequently, it has been demonstrated that the dual PI3-kinase/mTOR blocker NVP-BEZ235 can decrease cell growth of human neoplastic MCs [50]. Unfortunately, the results were not as satisfactory in ten SM patients with Everolimus, an oral mTOR inhibitor [51]. Moreover, mTORC1 and mTORC2 have been found to differentially regulate homeostasis in MCs. mTORC2 was demonstrated to be critical for the homeostasis of neoplastic and dividing immature MCs, while it played a minor role in the homeostasis of differentiated, nonproliferating, mature MCs. For this reason, targeting mTORC2 would represent a much more selective approach in the treatment of SM [52].

PKCδ is constitutively active in cells harboring the *KIT* D816V mutation when compared to wild-type cells where activation occurs after SCF stimulation [43]. Moreover, PKδ shows opposite functions in wild-type KIT cells compared to D816V KIT cells. In fact, it has been highlighted that the overexpression of PKCδ inhibits the growth of cells expressing wild-type KIT and enhances the growth of cells harboring the *KIT* D816V mutation [43]. For these reasons, PKCδ can be a therapeutic target in SM associated with mutations in the catalytic domain of *KIT*. Midostaurin also inhibits PKC.

It has also been observed that the SHP2 phosphatase is over-activated and over-expressed, and recruits PI3K and GAB2 leading to the activation of AKT/ERK pathways in *KIT* D816V mutant cells but not in *KIT* wild-type cells. Inhibition of SHP2 by the indole-based salicylic acid derivative II-BO8 repressed ligand-independent growth and survival of neoplastic MCs, and results were potentiated by combined PI3K inhibition [53].

Another molecule, FES, has been shown to play an essential role in downstream signaling in human and murine cells hosting the *KIT* D816V mutation. KIT D816V-dependent phosphorylation of FES is required for cell proliferation. FES itself is a tyrosine kinase and might, therefore, represent an attractive target [54]. 

Finally, the abnormal accumulation of neoplastic MCs in SM could result from the deregulation of BCL-2, BCL-xL, and MCL-1 pathways [55]. In fact, there is evidence of overexpression of BCL-2, BCL-xL, and MCL-1, which are anti-apoptotic molecules, in *KIT* D816V-positive neoplastic MCs in SM patients [56,57,58]. In contrast, a loss of expression of the pro-apoptotic BIM molecule in these cells has been detected [59]. The use of antisense oligonucleotides or RNA interfering against MCL-1 has led to reduced cell survival and increased apoptosis [59]. In a further study, it was found that a pan-BCL-2 family inhibitor (obatoclax) exerted growth inhibitory effects in cell lines and primary neoplastic MCs and synergized with dasatinib and midostaurin [60].

## 4. In Which Cell Type Does the D816V Mutation Arise? 

At least one-third of ISM patients and virtually all AdvSM patients show multilineage myeloid and/or lymphoid involvement of hematopoiesis by the KIT D816V mutation in BM cell compartments other than MCs (CD34 + hematopoietic stem and precursor cells, eosinophils, monocytes, and maturing neutrophils, and, to a lesser extent, also in T lymphocytes [7,61,62,63,64]. This indicates that the acquisition of the *KIT* D816V mutation may occur in an early pluripotent progenitor cell. Moreover, AHN cells from patients with SM-AHN are often found to harbor the KIT D816V mutation [62,65,66] supporting a common clonal origin for both disease components in a hematopoietic progenitor cell with multilineage potential. ISM patients with multilineage *KIT* D816V mutations have been shown to have a higher risk of progression to AdvSM or acute leukemia than patients with MC-restricted *KIT* D816V. The earlier the *KIT* mutation is hierarchically gained, the greater the resulting extent of hematopoietic involvement and the higher the risk of acquisition of additional secondary genetic ‘‘hits’‘ that may foster disease progression. Both the acquisition and the maintenance of these secondary alterations might be facilitated by the antiapoptotic survival pathways hyper-activated in neoplastic MCs because of the *KIT* D816V mutation. 

Moreover, among ISM cases showing multilineage involvement of BM cells by the *KIT* D816V mutation, half have been found to also have *KIT*-mutated BM-derived mesenchymal stem cells (MSCs), suggesting that in these patients the mutation arose in a common mesodermal ancestor of MSCs and hematopoietic progenitor cells. In line with this, Nemeth et al. reported that SM patients carry abnormal MSCs with slow proliferation, signs of senescence, and impaired osteogenic function in comparison to normal MSC colonies [67].

## 5. KIT Mutations: Diagnostic and Prognostic Considerations

In the case of a suspected SM, the diagnostic standard is mutation analysis of *KIT* in BM cells. If a BM aspirate is not available, the analysis can be performed on the marrow smear or from a paraffin-embedded biopsy sample [68]. In most SM patients, particularly ISM patients, the number of neoplastic MCs in the BM (MC burden) is very low. This sets the requirement for highly sensitive methods of mutation analysis. Several approaches have been reported in the literature. The following methods have been shown to detect the D816V mutation in 80% of SM patients and are therefore used for routine testing, although each of them has pros and cons (summarized in Table 1).

-RT-PCR combined with restriction fragment length polymorphism (RFLP): genetic analysis by RFLP is one of the most commonly used techniques for the identification of a known sequence variant. The RT–PCR followed by the RFLP assay represents a rapid, cheap, and easy method that allows the detection of *KIT* D816V mutation with a sensitivity of 0.05% [68,69]. However, it cannot reveal D816 variant mutations, nor can it detect mutations at other codons. Moreover, it does not allow for quantitation of the allele burden.-Nested RT-PCR followed by D-HPLC of PCR amplicons: this technique allows the identification of all the variants at codon 816. Sensitivity has been reported to be modest at 0.5–1% [70]. The method is relatively time-consuming and requires sample pooling for cost-effectiveness. Moreover, D-HPLC instruments are not widely available.-Peptide nucleic acid-mediated (PNA)-mediated PCR: this can be used for the detection of all the variants at codon 816 as well as at adjacent codons with a sensitivity of 0.1% [71]. Similar to the previous methods, PNA-mediated PCR is not quantitative. However, this is the recommended method for formalin-fixed, paraffin-embedded tissues [72].-Allele-specific oligonucleotide (ASO)-quantitative PCR (ASO-qPCR): this is one of the most sensitive methods for the identification and quantitation (allele burden) of the D816V mutation in different substrates. With ASO-qPCR it is possible to detect less than 0.01% *KIT* D816V mutation-positive cells [73,74,75]. Unfortunately, D816 variant mutations cannot be picked up by this assay.-Droplet digital PCR (ddPCR): a recent validation study has shown the potential of ddPCR to become the method of choice for D816V detection and quantitation because of its precision, accuracy, and sensitivity (0.01%) [76]. Moreover, ddPCR has proven capable to robustly quantitate the allele burden also in formalin-fixed, paraffin-embedded tissues [77].

It has to be noted that so far, no interlaboratory standardization initiative has been undertaken to compare different methods in terms of accuracy and reproducibility of mutation detection and quantitation of the allele burden, nor to validate and harmonize protocols. Such an initiative within a network of reference laboratories would be highly desirable.

ASO-qPCR and ddPCR have proven capable of detecting the D816V mutation even in PB [70,73,78,79,80]. Based on these studies, the European Competence Network on Mastocytosis (ECNM) suggested that PB analysis can be used for initial screening in patients with suspected SM, together with the measurement of serum tryptase levels [81]. If the serum tryptase level is >15 ng/mL and/or the ASO-qPCR detects the D816V mutation in PB cells, a BM examination is recommended, including *KIT* D816V mutation analysis by ASO-qPCR [82]. In the future, it will be interesting to assess whether another non-invasive approach like mutation analysis of cell-free DNA from plasma might prove a reliable alternative.

A small percentage (5–10%) of SM patients test negative for mutations at codon 816. In ISM cases, this may be due to the very low percentage of MCs infiltrating the BM. In such cases, fluorescence-activated cell sorting (FACS) or laser microdissection of MCs may help in enhancing sensitivity, but neither approach is readily feasible for routine diagnostic purposes. Additionally, some MCL cases have been reported to be D816V-negative by ASO-qPCR. Here, mutation status has mainly therapeutic implications, since the in case of regulatory-type mutations (or even wild-type *KIT*) imatinib would be a valuable option, while D816 variants would require midostaurin or non-TKI-based approaches such as cladribine or interferon. In such patients, a stepwise approach can be used, with PNA-mediated PCR to search for D816 variants in the first instance, and sequencing (Sanger sequencing or better next-generation sequencing (NGS) [83] of the entire *KIT* coding region in case no mutation whatsoever at codon 816 is identified. 

Quantitation of the allele burden has diagnostic and prognostic implications. Hoermann et al. reported that *KIT* D816V allele burden as assessed by ASO-qPCR correlated significantly with the WHO type of the disease: in ISM the median *KIT* D816V allele burden was 0.285% (range: 0.006–34.585%), in SSM 5.991% (range: 0.023–29.620%), in ASM 9.346% (range: 8.816–32.480%), and in SM-AHNMD 3.761% (range 0.083–50.183%)(*p* < 0.001) [78]. *KIT* D816V allele burden also correlated significantly with serum tryptase levels (*p* < 0.005) and age (*p* < 0.005). Moreover, a cut-off level of 2% was identified that allowed the definition of two prognostically distinct groups in terms of overall survival (OS). These findings have more recently been confirmed by Greiner et al. using ddPCR [76]. In this study, patients with advSM displayed a significantly higher *KIT* D816V allele burden (median, 2.43%) as compared to patients with ISM (median, 0.14%; *p* < 0.001). Moreover, ddPCR confirmed the prognostic value of an allele burden ≥ or <2 in defining groups with significantly different OS.

As mentioned above, detection of the *KIT* D816V mutation in hematopoietic cell compartments other than MCs (multilineage involvement) has been associated with the progression from ISM to AdvSM and a poor outcome. BM non-MC subpopulations to be analyzed for the presence of the *KIT* D816V mutation might be isolated via laser capture microdissection [64] or FACS [7], but both these approaches require expensive instrumentation and technical skills, which limits wide routine applicability. To overcome this problem, a comparison of the fraction of BM neoplastic MCs assessed by flow cytometry and of mutation-positive cells determined by ASO-qPCR may be used to indirectly infer multilineage involvement [74]. Moreover, Teodosio et al. have shown the utility of flow cytometric assessment of the immunophenotypic profile of BM MCs as a surrogate marker of multilineage involvement, since the aberrant expression of CD25 in association with a FcεRI^lo^, FSC^lo^, SSC^lo^, and CD45^lo^ immature phenotype of BMMC in the absence of coexisting BMMC with a normal phenotype, was associated with multilineage D816V *KIT* mutation, regardless of the diagnostic subtype of the disease [84]. However, the easiest and most straightforward approach has been suggested by Jara-Acevedo et al., who focused on the percentage of PB cells positive for the *KIT* D816V mutation by ASO-qPCR and proposed a cut-off of 6% above which we would be in the presence of multilineage involvement. An allele burden below 6% in ISM without skin lesions would be suggestive of MC-restricted *KIT* D816V mutation, while lineage involvement cannot be inferred from allele burden in the category of ISM patients with skin lesions [79]. 

A series of studies have suggested that the *KIT* D816V allele burden (or the expressed allele burden, that is, the allele burden quantitated at the transcript level [70]) may also prove useful for monitoring therapeutic efficacy in patients after cytoreductive therapy or allogeneic transplant [70,78] and more recently, in patients treated with midostaurin [85], at least in patients with advSM.

Since MCs are underrepresented in the liquid specimens, the mutation burden assessed in BM or PB probably underestimates the true tumor burden of the disease. Consistently with this hypothesis, a poor correlation between *KIT* D816V allele burden in BM or PB with MC infiltration or serum tryptase levels as surrogate markers of disease burden has always been found [70,78,79,86]. A recent study conducted by Greiner et al. used ddPCR to quantitate the D816V mutation in formalin-fixed, paraffin-embedded (FFPE) BM tissue sections (‘‘tissue mutation burden”) of a large series of SM patients [77]. The results of the study showed that allele burden in BM tissue sections is significantly higher and correlates better with MC infiltration in BM and serum tryptase levels. Moreover, the authors proposed that allele burden in FFPE should be further evaluated as a diagnostic criterion (B-finding) for SSM. Finally, a 9% cutoff of tissue mutation burden was identified that separated two subgroups with significantly different progression-free and OS. The tissue mutation burden remained an independent poor-risk marker in multivariate analysis taking B- and C-findings into account, in contrast to the liquid allele burden [77]. As a more direct marker of all residual *KIT* D816V positive cells, the tissue mutation burden measured by ddPCR might also become an important follow-up parameter for treatment response. Therefore, it was proposed that ddPCR-based measurement of *KIT* D816V mutation burden in FFPE BM tissue sections, as a new promising prognostic and minimal residual disease biomarker, should be included in all future studies of prognosis and therapy of SM [77].

**Table 1 ijms-21-03987-t001:** Advantages, disadvantages, and sensitivity of the different methods most commonly used for *KIT* gene mutation analysis.

Strategy	Pros	Cons	Lower Detection Limit	Ref
**PCR + RFLP (*Hinf* I)**	- Simple - Rapid - Reliable - Inexpensive	- Cannot reveal D816 variant mutations - Not quantitative	0.05%	Fritsche-Polanz, Br J Haematol 2001 [69]
**Nested RT-PCR + D-HPLC**	- Allows to identify all the variants at codon 816 or adjacent positions	- Not quantitative - Time-consuming - Needs dedicated, expensive instrumentation	0.5–1%	Erben, Ann Hematol 2014 [70]
**PNA-mediated PCR clamping + melting curve analysis**	- Allows to identify all the variants at codon 816 or adjacent positions - Works well also on FFPE tissues	- Not quantitative	0.1%	Sotlar, Am J Pathol 2003 [71]
**ASO-qPCR**	- Rapid - Relatively inexpensive - Quantitative	- Cannot reveal D816 variant mutations - Standardization and harmonization not yet undertaken	0.01%	Kristensen, J Mol Diagn 2011 [74]
**ddPCR**	- Simple - Rapid - Relatively inexpensive - Quantitative - Works well also on FFPE tissues	- Cannot reveal D816 variant mutations - Standardization and harmonization not yet undertaken	0.01%	Greiner, Clin Chem 2018 [76]

Abbreviations: PCR, polymerase chain reaction; RFLP, restriction fragment length polymorphism; RT, reverse transcription; D-HPLC, denaturing high-performance liquid chromatography; PNA, peptide nucleic acid; ASO-qPCR: allele-specific oligonucleotide-quantitative PCR; ddPCR: droplet digital PCR; FFPE, formalin-fixed paraffin-embedded.

## 6. Mutations in Genes Other Than KIT

Although activating mutations of *KIT*, especially D816V, play a crucial role in the pathogenesis of SM, the presence of the mutation alone is not sufficient to explain the spectrum of clinical phenotypes that characterize the disease. A study carried out by Zappulla et al. has shown that transgenic mice positive for *KIT* D816V develop, in most cases, a syndrome compatible with ISM but very rarely as aggressive as an ASM or MCL [87]. Therefore, in addition to the mutations that affect *KIT*, there must be further molecular events that contribute to the pathogenesis of AdvSM. The identification of these molecular alterations may help to refine prognostic models and highlight novel therapeutic targets to improve treatment approaches.

Indeed, mutations in genes, other than *KIT*, have been reported in >60% of patients with AdvSM, most frequently in cases of SM-AHN and to a lesser extent, in ASM patients. In contrast, very few SSM and ISM cases have been found to carry additional mutations [66,88,89,90]. These mutations, however, are not specific for SM but are also frequently found in other myeloid neoplasms like myelodysplastic syndromes, myeloproliferative neoplasms or, more rarely, acute myeloid leukemias [91,92,93,94]. They affect a series of genes encoding for signaling molecules (e.g., *CBL, JAK2, KRAS, NRAS*), transcription factors (e.g., *RUNX1*), epigenetic regulators (e.g., *ASXL1, DNMT3A, EZH2, TET2*) or splicing factors (e.g., *SRSF2, SF3B1, U2AF1*). In general, mutations in TET2, SRSF2, ASXL1, RUNX1, N/KRAS or IDH2 were most frequently identified in ASM-AHN and MCL-AHN, while mutations in JAK2, ETV6, U2AF1, EZH2, and SF3B1 were more often associated with ISM-AHN [95]. In order to investigate the hierarchical relationships of the clonal mutated populations, Jawhar et al. explored the mutation status of granulocyte–macrophage colony-forming progenitor cells (CFU-GM) in 19 *KIT* D816V+ patients with different types of SM. In this study, additional mutations (in *TET2, SRSF2, ASXL1, CBL, EZH2, U2AF1 SF3B1, JAK2, ETV6,* and *IDH2* genes) were only detected in SM-AHN patients. Coexisting *KIT* D816V and myeloid mutations in CFU-GM were observed mostly in ASM-AHN; in all these cases, *KIT* D816V was present only in a subset of colonies. The results of this study demonstrated that AdvSM with AHN is a multi-mutated stem cell neoplasm where the acquisition of a *KIT* D816V as a late genetic event drives phenotype modification towards SM, whereas ISM and SSM seem to not be (or only rarely) affected by mutations at the CFU-GM level [66].

Several groups have investigated the prognostic relevance of additional mutations in myeloid genes. Jawhar et al. were the first to show that the presence and number of mutated genes within the *SRSF2/ASXL1/RUNX1 (S/A/R)* panel was associated with adverse clinical and laboratory features (in terms of MC infiltration in the BM biopsy, serum tryptase levels, *KIT* D816V allele burden in PB, hemoglobin levels, platelet count, alkaline phosphatase and albumin levels, hepatomegaly with ascites, and weight loss) in a series of 70 patients with SM-AHN [96]. The number of C-findings was significantly higher in patients with two or more mutated genes as compared to patients with zero or one mutated genes, and the median OS was also significantly different among the three groups [70,95,97]. Assessment of the prognostic relevance of myeloid mutations by the Mayo Clinic group showed that the presence of *ASXL1* and/or *CBL* mutations [98], or the occurrence of ≥3 non-*KIT* mutations, were independently associated with significantly inferior OS [89]. The Spanish network later suggested that, in addition to that of the well-established *S/A/R* gene panel, somatic *EZH2* gene mutations might also provide prognostic information and proposed an *S/A/R/E* panel [88]. 

Jawhar et al. also evaluated the impact of mutations on response and progression of AdvSM patients treated with midostaurin [85]. Patients with mutations in the *S/A/R* panel had a significantly lower overall response rate and a significantly shorter OS as compared to *S/A/R*-negative patients. Acquisition of additional mutations in myeloid genes or an increase in the allele burden of pre-existing mutations marked disease progression [85].

## 7. Cytogenetic Abnormalities

Two recent studies have investigated the frequency of karyotype abnormalities in SM, with contrasting results. The German group examined the frequency and prognostic relevance of cytogenetic abnormalities in 109 patients with ISM (*n*  =  26) and AdvSM (*n*  =  83) with (*n*  =  73; 88%) or without AHN. Karyotypic abnormalities were detected only in SM-AHN (22%) patients. Seventy-five percent of these patients had myeloid mutations additional to *KIT* D816V. When patients were classified according to the type of karyotype abnormality into a good-risk group (*n* = 73; normal or favorable karyotype: del(5q); trisomy 8; del(1q); del(12p)) and a poor-risk group (*n* = 10; complex karyotype (defined as ≥3 abnormalities); monosomy 7; del(5q)), the latter displayed a significantly shorter OS, regardless of mutation status [99].

The Mayo Clinic group performed a similar analysis in 348 patients with SM (ISM, *n* = 142; AdvSM, *n* = 206; of whom 105 had AHN). The overall incidence of cytogenetic abnormalities was 15% (6% in patients with ISM/SSM; 26% in patients with SM-AHN; 8% in patients with ASM). No correlation between abnormal karyotype and “adverse” mutations could be found. An abnormal karyotype was associated with inferior OS in univariate analysis but not in multivariate analysis [100].

## 8. New Clinical and Clinico-Molecular Prognostic Scoring Systems in SM 

For a long time, the WHO classification has remained the strongest prognosticator [5]. However, over the years, a number of hematologic (anemia, thrombocytopenia, eosinophilia, excess of BM blasts.) [5,101], serologic (hypoalbuminemia; elevated alkaline phosphatase) [5,96], immunophenotypic (extent of myelodysplasia) [102], immunohistochemical (expression of CD123) [103] and clinical (organomegaly) [95] parameters, together with age, have been reported to have some prognostic value. Moreover, the advent of high throughput sequencing has led to the identification of molecular parameters with a strong adverse influence on survival and progression. For this reason, in recent years, several working groups have proposed prognostic scoring systems that integrate clinical and molecular data (obtained by using NGS myeloid gene panels) in order to more accurately estimate the risk of disease progression, predict survival probability, and direct treatment (these are summarized in Table 2). The first attempts at combining clinical and molecular parameters were made by the Germans and by the Mayo Clinic group shortly after the identification of recurrent myeloid mutations in AdvSM. Jawhar et al., based on the results of multivariate analysis, identified splenomegaly, with elevated alkaline phosphatase and mutations in the *S/A/R* gene panel as the key factors whose combination allowed the definition of two prognostically distinct subgroups (intermediate risk: 0–1 factors; high-risk: 2 factors) within AdvSM patients. Pardanani et al. proposed a ‘‘mutation-augmented’‘ clinical prognostic model for survival wherein a series of parameters identified by multivariate analysis concurred to determine survival probability of AdvSM patients: platelets < 150 × 10^9^/L (2 points), age > 60 years (1.5 points), serum albumin < 3.5 g/dL (1.5 points), presence of an ASXL1 mutation (1.5 points), and hemoglobin < 10 g/dL or transfusion dependence (1 point). Three distinct groups (low risk: score 0–1.5; intermediate risk: score 2–4.5; high-risk: score 5 or greater) with significantly different OS could be separated. These attempts represented the “embryos” of a series of recently developed prognostic scoring systems that are described below.

The Spanish Network on Mastocytosis (REMA) studied 322 ISM patients and identified by multivariate analysis serum β2-microglobulin levels > 2.5 μg/mL, *KIT* D816V allele burden >1% in BM and mutations in *ASXL1, RUNX1,* and/or *DNMT3A (A/R/D)* genes (with variant allele frequency ≥30%) as the best independent predictors for progression-free survival (PFS), whereas *A/R/D* mutations with variant allele frequency ≥30% were the only independent predictors for OS [104]. Two scoring systems were thus derived where variables were assigned one point each, that could successfully separate ISM patients with significantly different PFS and OS, respectively. However, they will require validation in an independent cohort of ISM patients [104]. 

The mutation-adjusted risk score (MARS), in contrast, was restricted to AdvSM and was developed in a series of 383 patients with available clinical and molecular information from the German Registry on Disorders of Eosinophils and Mast Cells (training set; *n* = 231) and in an independent cohort of patients from several ECNM centers in the United States and Europe (validation set; *n* = 152) [105]. MARS classified patients with AdvSM into three risk groups (low, intermediate, and high) with significantly different leukemia-free survival and OS based on four parameters, identified by multivariate analysis, that were assigned hazard ratio-weighted points: age (> or <60 years), hemoglobin levels (< or >10 g/dL), platelet count (< or >100 × 10^9^/L), and presence and number of high-risk mutations in the S/A/R gene panel (1 or ≥2) [105].

The Mayo Alliance Prognostic System (MAPS) was developed in a series of 580 patients seen at the Mayo Clinic between 1968 and 2015. Two separate but complementary risk models were elaborated. A clinical model based on five risk factors (age > 60 years, WHO-defined advanced SM vs. ISM/SSM, platelet levels < 150 × 10^9^/L, anemia defined as hemoglobin levels below the normal reference range, and increased serum alkaline phosphatase) could stratify SM patients with different survival probabilities, directly and proportionally decreasing as the number of risk factors increased. A subset of 150 patients for whom NGS-derived mutation information available was then used to develop a hybrid clinical-molecular model where adverse mutations (e.g., *ASXL1*, *RUNX1*, and *NRAS*) replaced anemia. However, since adverse mutations were observed only in AdvSM patients, the clinical-molecular model is only applicable in this specific category of patients [106]. 

Subsequently, a WHO class-independent MAPS model was refined, where SM subtype was replaced by serum albumin levels < 3.5 g/dL [106]. In the context of the new risk model, limited additional prognostic information was provided by adverse mutations, which appeared to almost exclusively cluster in high and very-high risk groups [107]. The performance of both MAPS models was later validated in a real-life setting of patients followed at the University of Florence [102].

Finally, the International Prognostic Scoring System for Mastocytosis (IPSM) was elaborated in 1639 SM patients included in the ECNM registry between 1978 and 2017 [108]. Patients with non-AdvSM (*n* = 1380) could be stratified into three risk groups (low, intermediate 1, and intermediate 2) based on two simple parameters, i.e., age ≥ 60 years and alkaline phosphatase ≥ 100 U/L. Patients with AdvSM (*n* = 259) were divided into four risk categories based on age ≥60 years, tryptase level ≥125 ng/mL, leukocyte count ≥ 16 × 10^9^/L, hemoglobin ≤ 11 g/dL, platelet count ≤ 100× 10^9^/L, and lack of skin involvement. Within each subgroup, risk categories had significantly different outcomes both in terms of OS and in terms of PFS. The prognostic value of both scores was confirmed by an independent validation cohort consisting of 413 patients with non-AdvSM and 49 patients with AdvSM [108]. 

**Table 2 ijms-21-03987-t002:** Variables included in the different prognostic scoring systems recently proposed for risk stratification of SM patients.

Variable		REMA [104]		* MARS [105]	* MAPS [106]	* IPSM [108]	
		ISM Only		AdvSM Only	All	Non-AdvSM	AdvSM
		PFS	OS	LFS, OS	OS	PFS, OS	PFS, OS
WHO class					+		
Age	≥60 years			+	+	+	+
Anemia	≤10 g/dL			+			
	≤11 g/dL						+
Thrombocytopenia	<100 × 10^9^/L			+			+
	<150 × 10^9^/L				+		
Leukocytosis	≥16 × 10^9^/L						+
Serum tryptase	≥125 ng/mL						+
Serum β2m	>2.5 μg/mL	+					
Serum ALP	>100 U/L					+	
	>ULN				+		
Mutations	*KIT* D816V in BM	+ (VAF ≥ 1%)					
	Myeloid mutations (NGS)	+ A/R/D (VAF ≥ 30%)	+ A/R/D (VAF ≥ 30%)	+ (S/A/R)	+ (A/R/NRAS)		

Abbreviations: REMA, Spanish Network for Mastocytosis; MARS, mutation-adjusted risk score; MAPS, Mayo Alliance Prognostic System; IPSM, International Prognostic Scoring System for Mastocytosis; ISM, indolent systemic mastocytosis; AdvSM, advanced systemic mastocytosis; OS, overall survival; LFS, leukemia-free survival; PFS, progression-free survival; WHO, World Health Organization; β2m, β2 microglobulin; ALP, Alkaline Phosphatase; ULN, upper limit of normal; BM, bone marrow; NGS, Next-Generation Sequencing; VAF, variant allele frequency; A, *ASXL1*; R, *RUNX1*; D, *DNMT3A*; S, *SRSF2*. *: score confirmed in an independent validation cohort.

## 9. Novel Molecular Alterations: SETD2 Non-Genomic Loss of Function in AdvSM

It has recently been reported that advSM is characterized by the loss of function of the *SETD2* tumor suppressor gene [109]. The *SETD2* gene, located on 3p21.31, encodes the only methyltransferase responsible for histone H3 lysine 36 trimethylation (H3K36Me3) [110]. H3K36Me3, a histone marker highly conserved from yeasts to humans, is enriched in the bodies of actively transcribed genes and tends to accumulate in exons rather than in introns. During transcription elongation, the trimethylation of H3K36 by SETD2 complexed with the phosphorylated C-terminal domain of RNA polymerase II restores a repressed chromatin structure to prevent spurious transcription from cryptic intragenic promoters [111]. Moreover, SETD2 is known to regulate alternative splicing by interacting with the large subunit of the heterogeneous nuclear ribonucleoprotein [112]. In addition to preserving transcription and splicing fidelity, other roles of SETD2/H3K36Me3 have more recently been hypothesized based on experimental observations in specific solid tumors (most frequently clear cell renal cell carcinomas (ccRCC)) and/or cell line models. H3K36me3 seems to play a key role in maintaining genomic integrity by modulating several pathways: i) H3K36Me3 is recognized by MSH6 and initiates the DNA mismatch repair pathway that corrects base-base and small indel mispairs generated during DNA replication [113] and ii) H3K36Me3 seems to dictate the choice of homologous recombination repair over the more error-prone micro-homology mediated end-joining at DNA double-strand breaks [114]. More recently, some SETD2 nonhistone substrates/interactors have been uncovered that play important roles in cellular homeostasis. SETD2 may bind p53 and seems to be necessary for its activation [111]. SETD2 loss of function might thus provide an alternative mechanism for evasion of the p53-mediated checkpoint in TP53-wild type cells. SETD2 trimethylates alpha tubulin during mitosis, ensuring the fidelity of chromosomal segregation and cytokinesis [115]. Furthermore, SETD2 depletion/mutation leads to decreased recruitment of DNA replication machinery and replication fork instability [116]. Figure 2 shows the schematic overview of the multiple known roles of SETD2.

Loss of function of the *SETD2* tumor suppressor gene has been reported in a wide variety of solid tumors and, more recently, in hematologic malignancies of both myeloid and lymphoid origin [117]. In ccRCC, *SETD2* loss of function has been associated with monoallelic deletions or copy-neutral loss of heterozygosity at chromosome 3p and mutation of the remaining allele. In acute leukemias, biallelic missense or truncating mutations can be found. In SM, a new post-translational mechanism of *SETD2* loss of function has been observed [109]. While biallelic truncating mutations were observed in one MCL case only, reduced SETD2 protein expression, and a parallel decrease in H3K36Me3 levels, were found in all the remaining 57 SM patients screened by Western blotting followed by densitometric analysis. The extent of SETD2 downmodulation correlated with the aggressiveness of the disease (ISM < ASM < MCL). SETD2 loss of function was also observed in the HMC-1 cell line, established from an MCL patient. In the absence of detectable genetic or epigenetic mechanisms like point mutations, gene deletion or promoter methylation, SETD2 loss of function was ascribed to enhanced protein degradation as a result of altered post-translational modifications. In fact, proteasome inhibition by bortezomib restored both SETD2 expression and H3K36Me3. Moreover, immunoprecipitation and immunoblotting experiments showed that inhibition of proteasome-mediated degradation resulted in an accumulation of SETD2 in its hyper-sumoylated and hyper-ubiquitinated form both in the HMC-1 cell line and in neoplastic MCs of AdvSM patients null for protein expression. Finally, as a fully reversible non-genetic event, SETD2 loss of function in AdvSM was suggested to be a novel promising druggable target: bortezomib treatment at subnanomolar concentrations proved remarkably effective in inducing apoptosis and reducing clonogenic growth of HMC-1 cells and neoplastic MCs from AdvSM patients [109]. 

In acute myeloid leukemias, *SETD2* is known to cooperate with a variety of major chromosomal or genetic aberrations that are the drivers of leukemogenesis [118]. Indeed, *SETD2* knockdown has been found to contribute to leukemia initiation and to accelerate disease progression and was associated with increased expression of mTOR and JAK-STAT pathway components, which are also KIT downstream effectors. Thus, although *KIT* D816V has a central role in SM pathogenesis, it can be hypothesized that SETD2 and H3K36Me3 deficiency may cooperate with, and potentiate the effects of KIT constitutive activation to determine the phenotype of AdvSM. However, besides the pathogenetic implications, SETD2 non-genomic loss of function may turn to harbor prognostic relevance, as it has recently been observed in several solid and hematologic tumors [119,120,121,122]. In our study, in a relatively small series of patients (*n* = 57) there was indeed a trend towards a shorter OS in SETD2/H3K36Me3-deficient patients [109]. Expression levels of SETD2 and H3K36Me3, assessed using WB or immunohistochemistry techniques, may turn out to be an informative approach capable of predicting the progression of ISM or SSM to AdvSM or reduced OS. Further studies with the integration of clinical and molecular data are warranted. 

## 10. Conclusions and Future Perspectives

The last few years have been characterized by the broad application of new advanced molecular technologies that have allowed a better understanding of the pathogenesis of SM. From a clinical point of view, these significant molecular advancements have refined diagnostication and prognostication and have opened new lines of research aimed at identifying further therapeutic targets. 

Highly sensitive and specific assays for the detection and quantitation of the allele burden of *KIT* D816V are now available. This will improve diagnostic accuracy, enabling a more straightforward analysis of FFPE specimens, as well as the move towards non-invasive testing in PB or, in the future, in cell-free DNA. The allele burden has been shown to have diagnostic and prognostic implications and might also prove useful for monitoring therapeutic efficacy, especially in cases of AdvSM treated with midostaurin or novel KIT inhibitors in clinical development like avapritinib or ripretinib. However, multicentric standardization efforts will be required in order to harmonize protocols and check for reproducibility. 

In addition to the progress in the genetic profiling of *KIT*, the use of NGS panels has outlined the genetic landscape that underlies the clinical heterogeneity of SM variants. The complementary use of highly sensitive and quantitative techniques for the mutational analysis of *KIT* and NGS panels for the characterization of associated gene mutations enables the dissection of patients into three key subgroups: (i) patients with MC-restricted *KIT* D816V (includes the majority of ISM characterized by non-progressive disease and a few slowly progressing AdvSM); (ii) patients with multilineage *KIT* D816V-involvement (includes the majority of AdvSM, a few progressing cases of ISM, and SSM); (iii) patients with “multi-mutated disease” (AdvSM, especially SM-AHN). Thanks to molecular discoveries, 2019 has witnessed an “explosion” of new prognostic scoring systems that combine clinical and molecular data. It will now be important to test further their ability to stratify more accurately patients with SM and to guide the correct choice of treatment in the context of prospective clinical studies, possibly conducted—given the rarity of the disease—through international cooperative efforts. 

A recurrent molecular alteration acting at the non-genetic level has also been recently identified in AdvSM, i.e., post-translational loss of function of the *SETD2* tumor suppressor. Assessment of levels of the SETD2 protein and of H3K36Me3 might provide prognostic information and should be further explored in a larger series of patients. Moreover, with SETD2/H3K36Me3 deficiency being a druggable lesion, it may represent a promising target in those AdvSM patients that fail KIT inhibitor therapy. Further preclinical and clinical evaluation of drugs like proteasome inhibitors, that are already approved for other indications could thus be easily repurposed.

Further expansion of our understanding of the underlying genetic and nongenetic mechanisms of SM will continue to impact on better patient management and ultimately improve therapeutic outcomes.

## Figures and Tables

**Figure 1 ijms-21-03987-f001:**
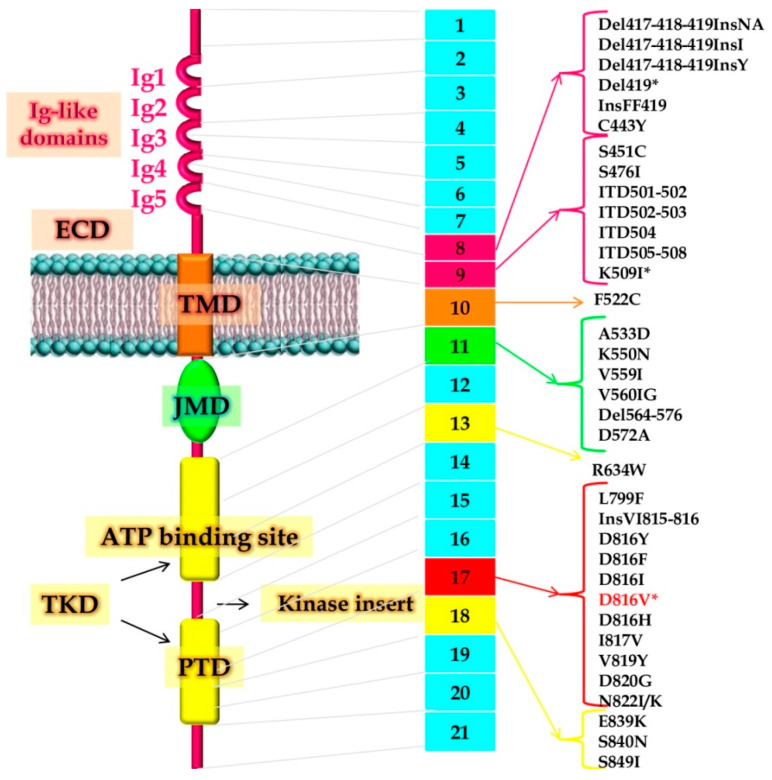
Representation of the structure of the KIT receptor, illustrating the known function of its domains and localization of all reported KIT mutations in adult patients with mastocytosis. On the left is shown the structure of the receptor. In the center, the 21 *KIT* exons and the most frequently identified mutations are reported. Abbreviations: Del, deletion; ECD, extracellular domain; Ins, insertion; ITD, internal tandem duplication; JMD, juxtamembrane domain; TKD, tyrosine kinase domain; PTD, phosphotransferase domain; TMD, transmembrane domain. *: mutation found in around 30% of pediatric patients and in > 80% of all adult patients.

**Figure 2 ijms-21-03987-f002:**
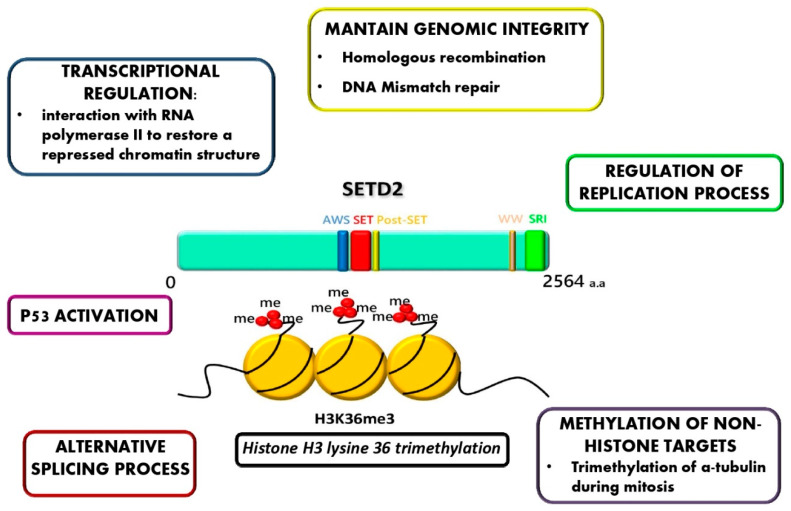
Schematic overview of the localization of the key functional domains of the SETD2 protein and its known roles. SETD2-mediated H3K36 tri-methylation is implicated in several cellular processes including: transcriptional regulation, genomic integrity, regulation of replication, methylation of non-histone targets, alternative splicing, and p53 activation.

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
