# Peer review of "Recent Advances in the Molecular Biology of Systemic Mastocytosis: Implications for Diagnosis, Prognosis, and Therapy"

_ijms, 2020, doi:10.3390/ijms21113987_

Round 1

Reviewer 1 Report

It's a competetive and well written review regarding the impact of genetic aberrations in patients with systemic mastocytosis. I have only one minor comment:

Section 7 (New Clinical and Clinico-Molecular Prognostic Scoring Systems in SM) + Table 2. Some of the prognostic systems (e.g. REMA) are not independently validated - please point out this in the text and in Table 2.

Author Response

We thank the Reviewers for their time and their careful evaluation of our manuscript. We have revised the text according to their helpful comments. In the revised version of our review, the changes have been highlighted using the "Track Changes" function. A detailed point-by-point reply can be found below.

Re to Reviewer 1

"It's a competetive and well written review regarding the impact of genetic aberrations in patients with systemic mastocytosis. I have only one minor comment: Section 7 (New Clinical and Clinico-Molecular Prognostic Scoring Systems in SM) + Table 2. Some of the prognostic systems (e.g. REMA) are not independently validated - please point out this in the text and in Table 2.’’

We thank this Reviewer for having pointed out this issue. As suggested, we have detailed in the text and in the table which prognostic systems were  independently validated and which were not.

Re to Reviewer 2

"This is a comprehensive and complete review on the present knowledge regarding the molecular biology of mastocytosis, with special attention paid to:

1) the various technics used to detect and quantify the KIT D816V mutant, their advantages and disadvantages, and implication for diagnosis, prognostication and follow-up,

2) the additional genetic defects found in advanced mastocytosis,

3) the new prognostication scoring systems,

4) A particular molecular alteration found in advanced SM; i.e. SETD2 loss of function and its implication for predicting disease evolution.

All in all the manuscript is well written and summarizes the most recent knowledge on the subject, the originality relying particularly on the paragraph on SETD2.

I have one major concern with this review: for a review whose ambituion is to deal with the molecular biology of systemic mastocytosis, there is only description of the molecular alterations, and nothing regarding the molecular consequences of these alterations. Particularly, a paragraph dealing with the molecular signaling pathways evoked by the mutant KIT D816V, which are totally different from those evoked by the wild-type KIT receptor, is missing. This is of interest as KIT D816V induces a number of abnormal signaling pathways (STAT5, Lyn, Btk, Fes, etc) which may account for the aggressiveness of the disease and might represent new therapeutic targets. This paragraph is, in my opinion, mandatory to be added here."

We do agree with this Reviewer that a paragraph should be added for better understanding of the molecular pathogenesis of mastocytosis. Thus, we have inserted a new section entitled ''Mutant KIT D816V receptor induces abnormal signaling pathways''.

Re to Reviewer 3

"The manuscript is a comprehensive review of the molecular diagnostics and recent aspects of molecular diagnostics in systemtic mastocytosis. The manuscript is well written and includes the recent publications in the field. I recommend publication of the manuscript."

We thank this Reviewer for his/her positive comments.

Reviewer 2 Report

This is a comprehensive and complete review on the present knowledge regarding the molecular biology of mastocytosis, with special attention paid to:

1) the various technics used to detect and quantify the KIT D816V mutant, their advantages and disadvantages, and implication for diagnosis, prognostication and follow-up,

2) the additional genetic defects found in advanced mastocytosis,

3) the new prognostication scoring systems,

4) A particular molecular alteration found in advanced SM; i.e. SETD2 loss of function and its implication for predicting disease evolution.

All in all the manuscript is well written and summarizes the most recent knowledge on the subject, the originality relying particularly on the paragraph on SETD2.

I have one major concern with this review: for a review whose ambituion is to deal with the molecular biology of systemic mastocytosis, there is only description of the molecular alterations, and nothing regarding the molecular consequences of these alterations. Particularly, a paragraph dealing with the molecular signaling pathways evoked by the mutant KIT D816V, which are totally different from those evoked by the wild-type KIT receptor, is missing. This is of interest as KIT D816V induces a number of abnormal signaling pathways (STAT5, Lyn, Btk, Fes, etc) which may account for the aggressiveness of the disease and might represent new therapeutic targets. This paragraph is, in my opinion, mandatory to be added here.

Author Response

(The authors gave the same response as above.)

Reviewer 3 Report

The manuscript is a comprehensive review of the molecular diagnostics and recent aspects of molecular diagnostics in systemtic mastocytosis. The manuscript is well written and includes the recent publications in the field. I recommend publication of the manuscript. 

Author Response

Thank you so much for your approval.

Best regards

Dr. Margherita Martelli

Round 2

Reviewer 2 Report

None